# Smelly communication between haemaphysalis longicornis and infected hosts with indolic odorants: A case from severe fever with thrombocytopenia syndrome virus

Zhitong Liu[1,2], Hao Feng[1], Xiaohe Liu[1], Bin Wu[2], Hong Zhang[3], Yi Sun[1]*, Jiahong Wu[4]*, Chunxiao Li[1], Jiafu Jiang[1]

1 State Key Laboratory of Pathogen and Biosecurity, Academy of Military Medical Sciences, Beijing, China, 2 Inspection and Testing Department of Jinhua center for disease prevention and Control, Jinhua, Zhejiang, China, 3 Military Hospital of 95948 Troops of PLA, Jiuquan, Gansu, China, 4 Basic medical School, Guizhou Medical University, Guiyang, Guizhou, China

* 378176938@qq.com (YS); jiahongw@gmc.edu.cn (JW)

## Abstracts

### Objects

Vector ticks' perception of characteristic odors emitted by infected hosts is key to understand tick's foraging behavior for infected host and design odor-based control strategies for tick-borne diseases.

### Methods

Laboratory mice knocked out for type I interferon (IFN) receptors (Ifnar$^{-/-}$) were used to develop a simulated host by intraperitoneal infection with *Bandavirus dabieense* (SFTSV). Urine and fecal samples were collected 4 days post-infection and analyzed to detect differential volatile metabolites (DVMs) during infection. Next, the two salient odor cues among the SFTSV-induced host DVMs, indole and 3-methylindole, were used to test the olfactory response of *Haemaphysalis. longicornis* by electroantenno-graphic detection (EAD) and Y-tube olfactometry, respectively. To gain insight into the potential olfactory mechanism, two olfactory-associated proteins, Niemann-Pick type C2 (NPC2) and Odor Binding Protein-like (OBPL) proteins were annotated from the transcriptomic data derived from *H. longicornis* forelegs. Online tools were used to predict the ligand binding properties of the two proteins to the two indole candidates. Simultaneously, quantitative RT-PCR using β-actin as an internal reference gene was used to monitor the relative transcript levels of NPC2 and OBPL proteins under the stimulation of two indole candidates. The significantly regulated proteins were cloned and expressed with the vector plasmid pET-28b *in vitro*. The purified proteins were tested for the binding properties to the two indole candidates.

**Data availability statement:** The olfactory related proteins OBPL and NPC2 were annotated from H. longicornis were deposited to NCBI under the accession no. PV029724 (NPC2) and no. PV029725 (OBPL) of SRR 28426262 in the BioProject PRJNA 1091141: HaeLTT.

**Funding:** The work was supported by the National Key Research and Development Program of the People's Republic of China (2022YFC2305001 to Y.S.) and the National Science and Technology Fundamental Resource Investigation Program of the People's Republic of China (2022FY100900 to Y.S.). The funders had no role in study design, data collection and analysis, decision to publish, or preparation of the manuscript.

**Competing interests:** The authors have declared that no competing interests exist.

## Results

SFTSV-infected Ifnar$^{-/-}$ mice upregulated 11 DVMs in fecal samples, mostly indoles and phenols, along with indole biosynthesis and related metabolic processes. In the urine samples, 29 DVMs were downregulated in the infected host, with eucalyptol and phenylalanine acid being the most altered. We test the olfactory responses of *H. longicornis* to indole and 3-methylindole, which influence tick foraging behavior. The olfactometers showed that the tick preferred both indole and 3-methylindole. EAD tests showed that stimulation of the olfactory receptor neuron in Haller's organ produced significant active potential in response to indoles. Two olfactory proteins, NPC2 and OBPL, were successfully annotated from *H. longicornis* foreleg transcriptomic data. NPC2 has a β-barrel structure that binds signal chemicals, while OBPL is a classical OBP with a hydrophobic binding cavity. When monitoring the transcript levels of NPC2 and OBPL in the tick forelegs, the increased transcript level (1.2-1.4 folds change) of OBPL was observed following indoles stimulation, compared to the downregulated level (0.6-0.8 folds change) of NPC2 under the same circumstances. The *OBPL* and *NPC2* gene from *H. longicornis* were successfully cloned and expressed as inclusion proteins respectively. The purified OBPL (20.28 kDa) showed higher affinity for both indole (Ki 2.256μM) and 3-methylindole (Ki 4.191μM) than NPC2 in the competitive fluorescence binding assays with 1-NPN as a competitor.

## Conclusions

Facilitated by the olfactory OBPL protein in Haller's organ, *H. longicornis* smells and is attracted to the characteristic indolic scents of hosts induced by SFTSV infection. Olfactory associations between infected hosts and vector arthropods could provide a new perspective to understand host foraging behavior and design novel control strategies for tick-borne diseases based on pathogen-induced scent according to chemical ecology theory.

### Authors' summary

The olfactory perception of characteristic scent emitted by infected hosts is a critical component in comprehending the ticks' foraging behavior, and consequently, formulating odor-based methodologies to combat tick-borne diseases. In this study, two salient infection scents, indole and 3-methylindole, were identified from the Ifnar$^{-/-}$ mice infected by *Bandavirus dabieense* (SFTSV) with differential volatile metabolite assays. The two indolic scents were then validated for their olfactory response of *H. longicornis* by a combination of field studies, laboratory experiments, and molecular analyses. We conclude that *H. longicornis* smell the characteristic scents from the SFTSV-infected hosts using its unique OBPL protein in the olfactory apparatus (Haller's organ) and adjusts the tick's foraging behavior to

target infected subjects. The findings of this study offer insights into the intricate interactions between ticks, their hosts, and pathogens, which could inform novel strategies for the prevention of tick-borne diseases such as SFTS.

## Introduction

Ticks are obligate, temporary ectoparasitic blood feeders on vertebrates and are the primary vector of diverse microbial agents among hematophagous arthropods. The ability to locate and detect vertebrate hosts is a crucial aspect of the tick's survival, as it provides a source of nutrition. Hard ticks typically employ two main strategies when locating vertebrate hosts. They are ambush strategists and host-hunting strategists [1]. The ambush strategists are able to detect cues indicative of the presence of a vertebrate host, including $CO_2$ [2], odorants [3,4], body heat [5], infrared radiation [6], and vibrations [7]. They then proceed to climb onto the host in order to locate a suitable feeding site for attachment. In contrast, the host-hunting strategists exhibit a pronounced tendency to actively seek out vertebrate hosts, responding to a wide range of cues derived from the hosts. Actually, most tick species may use one or both strategies, and often alter them during host-foraging periods depending on their life stages [1]. Consequently, the ticks of both strategies respond in a tactical and precise way to different olfactory cues from their respective hosts. During the host-foraging period, the primary sense employed by ticks for locating hosts is olfaction, utilizing chemoreceptors situated on the Haller's organs on the foreleg tarsi [8]. These receptors allow ticks to detect a wide range of volatile organic compounds (VOCs), which then inform appetitive behaviors, such as questing and movement towards vertebrate hosts [9]. At present, a number of VOCs have been identified in a range of biological samples, including host dermal pelage [9], breath (e.g., $CO_2$, and 1-octen-3-ol) [2,3], secretions [10] from diverse glands [4], urine [3], and feces [11]. With the increasing VOCs identified, endotrophic or ectotrophic microbiota of host animals has been the subject of considerable attention. These microbes produce and release diverse volatile metabolites that may function as host VOCs cues, thereby prompting synergistic attraction of competent vector arthropods to infected hosts, which in turn facilitates pathogen transmission. The majority of ectotrophic microbiota are currently found to be breeding on host skin, lipids, furs and diverse glands, creating a complex microenvironment that is favorable for the particular ectotrophic microbiota in question. This process is facilitated by the secretion of lipids, salts, enzymes, antimicrobial peptides and a plethora of other chemical compounds [12]. For example, the preference for *Propionibacterium* in sebaceous gland ecotones, *Staphylococcus*, *Corynebacterium*, *Aspergillus* and *Flavobacterium* in damp or dry environments are usually reported to produce diverse metabolites (acetophenone *etc.*) that may serve as host semiochemical cues for foraging arthropod vectors [13]. These metabolites include acids, fatty acid derivatives, aldehydes, and sulfur- and nitrogen-containing compounds [14,15], which are readily diffused and employed as informative VOCs to manipulate vectorial arachnids and insects due to their low molecular weight (< 300 Da) and high vapor pressure (0.01 kPa at 20°C). Consequently, the ectotrophic microbiota produces unique olfactory signals that alter the perception of host animals, thereby influencing their attractiveness to hematophagous arthropods [16,17]. Furthermore, endotrophic microorganisms of host animals may also exert a considerable influence on the foraging behavior of hematophagous vectors. The impact of diverse pathogens or endotrophic microbes on vector-borne disease control has been a significant area of concern within the context of a novel proposed control strategy. This strategy aims to reduce host attractiveness and, as a consequence, block the transmission of pathogens. Recent studies have yielded fascinating insights into the relationship between hosts infected with Dengue, Zika viruses and mosquito behavior [17]. The results indicate that viral infection inhibits RELM-α, a major antimicrobial peptide (AMP) in the host skin, and leads to a perturbation of the skin microbiota. This led to the release of specific host-produced VOCs, such as acetophenone, which increased mosquito attraction and facilitated virus transmission [17,18]. In addition to these viral agents, similar characteristic VOCs of infected hosts have also been evaluated in the presence of *Borrelia afzelii* [19] and *Anaplasma phagocytophilum* [11] in vector ticks, and for *Plasmodium falciparum* [20,21] in mosquitoes, and for *L. infantum* in sand flies *Lutzomyia longipalpis* [22], and for *Trypanosoma vivax* in tsetse flies *Glossina pallidipes* [23] respectively. Thus, these

characteristic host VOCs, induced by specific pathogens or endosymbionts, facilitate pathogen transmission by attracting more competent vector arthropods through the tripartite pathogen-host-vector interactions. For example, host VOCs induced by *P. falciparum* gametocyte parasitizing in erythrocytes, attracted 2.5 times more *Anopheles* spp. mosquitoes than non-induced groups [21]. Significantly different levels of VOCs compositions (*i.e.*, α-pinene and 3-carene) were detected in the breath of malaria victims [24,25]. The increased attractiveness of *Anopheles stephensi* mosquitoes to mice infected with *P. chabaudi* and a clear difference in the VOCs production profiles between infected *vs* non-infected mice [26], suggest that the cause-effect relationship between pathogens and altered host metabolites mediating the seeking behavior of arthropod vectors. As further examples, the heavy burden of immature *Ixodes ricinus* on *Myodes glareolus* infected with *Borrelia afzelii* was found [19] and *Ixodes hexagonus* on the *Anaplasma phagocytophilum* positive hedgehog *Erinaceus europaeus* [11] had been documented and explained by the increased levels of VOCs (aromatic heterocyclic compounds) in urine (mice) and the feces (hedgehog) of infected hosts [11]. Vector-borne pathogens alter infected hosts' VOC profiles to make them more attractive to blood-seeking vectors, increasing vector-borne disease transmission. Such an intriguing ecological hypothesis would be supported by more scientific evidence from sandflies [27], tsetse flies [23,28,29], tabanids [30], kissing bugs [31] and other hematophagous arthropods. However, more detailed mechanisms have yet to defined. In these circumstances, the distinctly different profiles of the volatile metabolites produced by endotrophic or ectotrophic microbes may result in a synergistic attraction that is greater than the sum of their individual effects. And the identification of characteristic VOCs produced by infected hosts would be a useful tool for the discrimination of infected hosts from non-infected hosts and for the diagnosis of vector-borne pathogens [32]. With diverse and dense olfactory receptors in their antennae, mosquito senses unique $CO_2$, $NH_3$ or other VOCs and guide themselves through different stages with oviposition or host foraging behaviors. Conceivably then, olfactory interactions between characteristic VOCs and specific odorant receptors or odorant binding proteins may also inform the tick's search tactics for a favorable host. Our epidemiological surveys for SFTSV in the field had shown a significantly higher burden of immature *H. longicornis* on the infected rodents, indicating *H. longicornis* was attracted to and parasitized on the SFTSV-positive hosts compared to the SFTSV-negative groups (S1 Fig). We therefore tested the hypothesis that *H. longicornis* is captured by or attracted to SFTSV-infected hosts and whether the attraction of *H. longicornis* is mediated by host odor induced by SFTSV infection.

## Methods and materials

### Ethics statement

Ethical approval was granted by Institutional Review Board in the Academy of Military Medical Sciences, Peoples Liberation Army P. R. China (AMMS-IRW-202000036). All experiments were performed in accordance with relevant guidelines and regulations.

**Bandavirus dabieense:** Severe fever with thrombocytopenia syndrome virus (SFTSV) Xinyang strain: (Accession no. JQ341188,JQ341189 and JQ341190 for N, S, L, directly submission): a virus strain isolated from a patient from Shangcheng County, Xinyang City, Henan Province, China.

**Ticks:** SFTSV-free *Haemaphysalis longicornis*: *H. longicornis* ticks originally collected from Changping District, Beijing, China, were maintained at the insectary of AMMS at 26°C with 85% humidity under a 12/12h light/dark photoperiod. All the tick individuals were harvested from SFTSV-free maternal adults and the subsequent offspring were sampled to verify the negative infection of SFTSV by the specific RT-PCR screening. Unfed nymphs were used in the experiments at the age between 10–20 days post-molt.

### *Ifnar-/-* mouse

Homozygous *Ifnar*[-/-] mice, female, aged 6–8 weeks, were purchased from Cyagen Biosciences Inc (Guangzhou, China). All experiments were performed using institutional guidelines for animal care and experimentation under the official approvals obtained prior to the *in vivo* assays.

## Surveillance of tick burden on *Apodemus agrarius* in SFTSV endemic area

*Apodemus agrarius* mice were trapped in Ningbo City, Zhejiang Province from 12 June, 2018–14 June, 2020 using per-forated Sherman traps (LFAHD folding trap, 7.62 × 8.89 × 22.86 cm, Sherman Traps, Inc, US) baited with peanut butter, sunflower seeds and apples. Traps were placed in the late afternoon and processed the following morning. The number of trap squares used (4 traps per square, with an inter-trap distance of 20m) depended on the sample size. We calculated the rodent abundance index per 100 traps as the sum of rodents over the trapping effort to control for the differences in trapping effort between sampling sites. Power analysis showed that a capture success of 10% to 25% at each sampling site was sufficient for evaluation. The captured rodents were delivered to the laboratory, and anesthetized in a closed container with ether or chloroform for about 10min to prevent the escape and bite of various parasites on the rodent body surface, then *A. agrarius* mice were morphologically identified according to the identification pictures of common medical vectors [33], as they were the most common rodent species and potential host reservoir of SFTSV in our sampling sites. After anesthesia with isoflurane, the mice were thoroughly checked for all ticks with fine forceps. Ticks harvested were counted and identified to species and life stage by morphological characteristics following the taxonomic key developed by Teng and Jiang [34]. The mice were also dissected to obtain liver, spleen and lung tissues which were preserved at liquid nitrogen until further processing.

## Experimental infected *Ifnar-/-* Mice with SFTSV

Female *Ifnar$^{-/-}$* mice (n = 6/group) were intraperitoneally injected with SFTSV (Xinyang strain) at a tissue culture infectious dose of 50 (TCID$_{50}$) per mouse, as these doses ensured mouse survival after challenge and would benefit subsequent sample collection and observations, an equivalent volume of PBS was used as a control [35]. Body weight changes, anal temperatures, and SFTSV RNA loads were monitored daily for 12 days immediately after SFTSV infection. SFTSV RNA loads in tail vein blood samples were determined by quantitative RT-PCR (qRT-PCR) as described below.

## Fecal and urine samples collected from SFTSV-infected *Ifnar-/-* mice

The fecal and urine samples were collected from *Ifnar$^{-/-}$* mice infected and noninfected with SFTSV. After SFTSV RNA loads > the initial infection dose in the challenged mouse, the mouse was individually placed into a sterilized Tecniplast E-Chiller cabinet (Tecniplast, China) to collect the fresh fecal and urine samples at 4°C. About 0.2g of fecal samples were taken in 1mL falcon tubes, homogenized and centrifuged (50,000 × g at 10°C for 2h), the supernatant is collected. The urine samples of 0.2mL in falcon tubes were centrifuged to remove any debris (50,000 × g at 10°C for 15min). At the time of solid phase microextraction (SPME) analysis, each vial of fecal water or urine samples was placed in 60°C for 5min, then exposed to 120µm DVB/CWR/PDMS fiber (Agilent) for 15min at 60°C.

## GC-MS analysis for differential volatile metabolites (DVMs) in infected *Ifnar-/-* mice

After sampling, desorption of the VOCs from the fiber coating was carried out in the injection port of the GC apparatus (Model 8890; Agilent) at 250°C for 5 min in the splitless mode. The identification and quantification of VOCs was carried out using an Agilent Model 8890 GC and a 7000D mass spectrometer (Agilent), equipped with a 30m × 0.25mm × 0.25µm DB-5MS (5% phenyl-polymethylsiloxane) capillary column. Helium was used as the carrier gas at a linear velocity of 1.2mL/min. Temperatures for inlets and MS source were taken as 250°C and 230°C, respectively. The oven temperature was programmed from 40°C (3.5min), increasing at 10°C/min to 100°C, at 7°C/min to 180°C, at 25°C/min to 280°C, hold for 5min. Mass spectra were recorded in electron impact (EI) ionization mode at 70eV. The quadrupole mass detector, ion source and transfer line temperatures were set at 150, 230 and 280°C respectively. The ion monitoring (SIM) mode of the MS was selected for the identification and quantification of the analytes. Briefly, the total ion chromatogram was obtained and then the mass spectra were identified. The detected metabolite peaks were identified using the NIST08 (National

Institute of Standards and Technology) mass spectral library, and the extracted compounds were aligned and normalized to the internal standard ribityl [36]. (1) Principal component analysis (PCA). Unsupervised PCA was performed using the prcomp statistical function within R (www.r-project.org). The data were unit variance scaled prior to unsupervised PCA. (2) Hierarchical cluster analysis (HCA) and Pearson correlation coefficients (PCC). The HCA results of samples and metabolites were presented as heatmaps with dendrograms, while the PCC between samples was calculated by the corfunction in R. Both HCA and PCC were carried out by R package Complex-Heatmap. For HCA, normalized signal intensities of metabolites (unit variance scaling) are visualized as a color spectrum. (3) Selected differential metabolites. For the paired group analysis, differential volatile metabolites (DVMs) were determined by variable importance in the projections (VIP) (VIP > 1) and absolute $Log_2$Fold Changes ($|Log_2FC|$) ≥ 1.0. VIP values were extracted from the partial least squares discriminant analysis (PLS-DA) results, including score plots and permutation plots, generated using the R package MetaboAnalystR [37]. The data were logistically transformed (log2) and mean centroided prior to PLS-DA. A permutation test (200 permutations) was performed to avoid overfitting. (4) KEGG Annotation and Enrichment Analysis. Identified metabolites were annotated using the KEGG compound database (http://www.kegg.jp/kegg/compound/), the annotated metabolites were subsequently mapped to the KEGG pathway database (http://www.kegg.jp/kegg/pathway.html). Pathways with significantly regulated metabolites mapped were then fed into MSEA (metabolite sets enrichment analysis), their significance was determined by hypergeometric test *P*-values.

## SFTSV screening

SFTSV was tested in samples of live ticks and mouse tissues (liver, spleen and lung). Samples were homogenized and centrifuged. Total nucleic acids (including RNA and DNA) were extracted from the homogenates using a Viral RNA MiniKit (52904) (QIAGEN, China) according to the manufacturer's instructions. Real-time fluorescence quantitative PCR assays for the target genes were then performed using the Takara one-step RT-PCR kit (Takara, Japan) [38]. The total volume of the reaction system of SFTSV was 25 µL, including 2 × reaction mix 12.5 µL, probe (10 µM) 0.3 µL, upstream and downstream primers (S2 Table) (10 µM) 0.5 µL, RNA template 5 µL, enzyme mix 1.0 µL, and deionized water supplement. The cycling parameters were 50°C for 30min (one cycle), 95°C for 10min (one cycle), 95°C for 15s and 60°C for 45s (40 cycles). Cycle threshold values of SFTSV was set as ≤35. Data were analyzed using the software supplied by the manufacturer.

## Choice tests

*H. longicornis* nymphs were allowed to choose between Y-tube olfactometer [1] arms by random assignment, with one arm containing SFTSV-positive *Ifnar^{-/-}* mice, healthy mice or DVM candidates (indole or 3-methylindole), and the other arm as an air control. The flow of odor-laden air was maintained at a rate of 0.2L/min for 2 min before the tests began. Each tick that proceeded 8.0 cm or more into either a treatment or control side arm as its first choice within 12min was considered a responder. All other ticks were considered non-responders.

## Potent electrophysiological responses of *H. longicornis* to volatile indole compounds

*H. longicornis* nymphs were subjected to test the olfactory response with EAD following Josek's protocol [39]. Briefly, the tick individual was attached ventrally to a circular metal plate (Ø 1 cm) with double-sided adhesive tape. Both the indifferent and the recording electrodes were filled with $10^{-2}$M KCl and 1% solution of polyvinylpyrolidone K90 (Fluka, Switzerland). The indifferent electrode, connected to the ground via a chloridized wire, was inserted in the region posterior to the scutum, after piercing it with a fine forceps. One of the forelegs was orientated to expose the anterior sensilla of Haller's organ and immobilized with an adhesive tape. Pedal nerves of the forelegs were destroyed by pinching the coxa with fine forceps to prevent muscle activity during electrophysiological recording. In order to improve contact, the tips of the distal knoll sensilla were cut with metal knives fitted on micromanipulators. Preparation of the ticks and subsequent imaging

were performed under visual control (Leica MZ12 stereomicroscope, 350 × magnification). Prior to imaging, the tip of the sensillum was cut with a piece of a razor blade in a holder mounted in a micromanipulator (NMN25, Narishige) under visual control. Recordings from the olfactory receptors were accomplished with glass electrodes connected to a high-input impedance preamplifier (10×) (Syntech INR-5, Hilversum) and brought into contact with the cut tip of the sensillum with the aid of micromanipulators. The recordings were sampled (13,714.3samples/s) and filtered (10–3000Hz, with 50/60Hz suppression) via USB-IDAC connection to a computer (Syntech, Hilversum). At least 10 ticks were tested with both indole and 3-methylindole at concentrations of 0.1, 1, 10 and 100µg/mL. The action potentials were extracted as digital spikes according to top–top amplitudes, using the Autospike software (version 3.9, 16 June, 2009, Syntech NL). The recording duration was 10s, and the stimulus was applied 500ms after the start of the recording. Responses were evaluated according to the difference in the number of spikes between the 500ms stimulation period and the 500ms period before the start of stimulation (dsf = difference in spike frequencies). One-way analysis of variance with repeated measurements (ANOVAR) followed by Bonferroni's post hoc correction test was used to analyze the mean dsf discharged at each concentration of each treatment (substance tested), using the SPSS 17.0 software package (SPSS Inc., Chicago, IL, USA). Statistical significance was achieved when $P < 0.05$. Odorants that elicited a significantly higher dsf than the blank stimulus were considered active. Two-way ANOVAR with Bonferroni's post hoc correction test was used to compare the effects of pairs of stimulating odorants. In cases where the sphericity assumption was violated, Greenhouse-Geisser (G-G) or Huynh-Feldt (H-F) corrections were applied when ε' > 0.75 or ε' < 0.75, respectively.

## Monitoring the transcript level of candidate olfactory proteins with qRT-PCR

The transcript profiles for OBPL and NPC2 during the induction of indole and 3-methylindole at different concentrations were measured by quantitative RT-PCR using an ABI 7500 Real-Time PCR System (Applied Biosystems, Carlsbad, CA, USA). The housekeeping gene, β-actin, was used to normalize target gene expression and to correct for sample-to-sample variation. TaqMan primers (Table 1) for the amplification of β-actin, NPC2 and OBPL were reported by Cui *et al.* (2022) [40]. For the qRT-PCR reaction, cDNA was diluted to 200ng/mL. Each reaction was performed in a 20µL final volume containing 10.0µL TB Green Premix Ex Taq (TaKaRa), 0.8µL of each primer (10mM), 0.5µL probe (10 mM), 0.4µL Rox Reference Dye II, 2µL sample cDNA (200ng) and 6.0µL sterilized $H_2O$. The reaction cycling parameters were as follows: 95°C for 30 s, then 40 cycles at 95°C for 5s and 60°C for 34 s. And then, the reaction of 95°C for 15s, 60°C for 1min and 95°C for 15s was added to establish the melting curve. For data reproducibility, reactions were performed in triplicate, and three biological replicates were assessed. Negative controls were non-template reactions (sterilized $H_2O$ instead of cDNA). According to the method described, the relative quantification of the target genes' transcripts between different treatments was calculated using the comparative 2^(-△△Ct) method [39] with thresholds of |$Log_2$Fold Change| ≥ 0.2, and $P$-value≤0.05 set significant differences. Comparative analyses of the transcript level of target genes amongst different treatments were conducted using a one-way nested analysis of variance followed by Tukey's honestly significant

**Table 1. The primer sequences used in the study.**

| Primers | Sequences |
|---|---|
| *H. longicornis* OBPL-F | GGAAAGACAACAGAAGGCCCTA |
| *H. longicornis* OBPL-R | TGAGGAGTTGATTGGTGCGCT |
| *H. longicornis* NPC2-F | TGGATTTTCACCGCTCTGCT |
| *H. longicornis* NPC2-R | CGAAATCGCCTCCGCATTTT |
| β-actin-F | CGTTCCTGGGTATGGAATCG |
| β-actin-R | TCCACGTCGCACTTCATGAT |

difference test. The relative mRNA transcript levels in nymphs' forelegs between treatments were compared using Student's t-test. All analyses were conducted using SPSS STATISTICS version 18.0 software (SPSS Inc., Chicago, IL, USA).

## Annotation of candidate olfactory proteins and their predicted 3D structures

Total foreleg RNA was isolated by trizol reagent (Invitrogen, Carlsbad, CA, USA) to construct cDNA library using the Creator SMART cDNA Library Construction Kit (Clontech, Mountain, CA, USA). The cDNA library was sequenced with Illumina sequencer. Candidate OBPL and NPC2 genes were identified by BlastX and MotifSearch program. OBPL and NPC2 protein sequences identified in *H. longicornis* and others reported in arthropod's species were aligned using ClustalX 1.83 respectively. The putative N-terminal signal peptides were predicted by the SignalP V5.0 program (http://www.cbs.dtu.dk/services/SignalP/) [41]. A Swiss model method [42] was used to search structural templates for the two candidate olfactory proteins. Several identified OBPs structures were used as templates to construct 3D structures for the two candidates, the one with the highest score of Profiles-3D was retained. The Profiles-3D method and Ramachandran plot were used to evaluate the rationality of the established 3D model [43]. Molecular docking was performed by the on-line program SWISSDOCK using default parameters [44], the docking scores of AC and SwissParam were used to evaluate the fitness of ligand and target proteins [45].

## Expression and purification of recombinant OBPL and NPC2 proteins

Two specific primer sets are designed to clone the coding region of OBPL/NPC2 in *H. longicornis* as followed: OBPL-F: 5'- GTCATATGGCTGCCACGTACACGTCC-3', OBPL-R: 5'-TGAAGCTTTCAGTGGCTTC CGGGCAA-3'; NPC2-F: 5'-AACATATGAAATACTACACGGATTG-3', NPC2-R: 5'- AAGAAT TCTTATTGTATCTTGGCGGC-3' (Underlined showed *Nde* I and *Hind* III enzyme sites in the forward and reverse primer, respectively.) The PCR products were cloned into the bacterial expression vector pET28b (Promega, Madison, WI) between the *Nde* I and *Hind* III restriction sites, and verified by sequencing (S6 Fig). Plasmid containing the correct insert was extracted and transformed into *Escherichia coli* BL21(DE3) competent cells. A verified single colony was grown overnight in 50mL LB broth (including 100mg/mL Kanamycin). Five liters of LB medium was inoculated with the 50mL overnight culture at 37°C for 2–3 hours until the absorbance at $OD_{600}$ reached 0.6. The proteins were then induced with isopropyl-b-D-thiogalactopryranoside (IPTG) with a final concentration of 1 mM at 37°C for 6h. The bacterial cells were harvested by centrifugation (8000g, 10min), resuspended in the lysis buffer (80mM Tris-HCl, 200mM NaCl, 1mM EDTA, 4% glycerol, pH7.2, 0.5mM PMSF), lysed by sonication (10sec, 5 passes) and centrifuged again (12000g, 10min). The soluble fraction and the whole pellet were analyzed by sodium dodecyl sulfate polyacrylamide gel electrophoresis (SDS-PAGE) and found the target proteins mainly present in the inclusion bodies. Insoluble proteins were washed with 0.2% triton X-100 in 50mM Tris buffer (pH6.8) and then dissolved in 6M guanidinium hydrochloride, the protein refolding protocols performed using the redox methods [46]. Soluble and refolded target proteins were purified with two rounds of Ni-21 ion affinity chromatography (GE Healthcare), and the His-tag was removed using recombinant enterokinase (Novagen, Madison, WI, USA) respectively. The highly purified protein was desalted through extensive dialysis, and the size and purity of the recombinant protein were confirmed by 15% SDS-PAGE. The concentration of the purified target proteins was measured by the Bradford method using BSA as standard protein [47].

## Fluorescence competitive binding assay

Both indole and 3-methylindole were selected for fluorescence competitive binding assays according to the DVMs with significant olfactory responses in SFTSV infected mice. This experiment was performed on an F-380 fluorescence spectrophotometer (Gangdong Sci & Tech Development. Co., Ltd, Tianjin, China) at room temperature (25°C) with a 1 cm light path quartz cuvette and 10-nm slits for excitation and emission. The excitation wavelength was 337nm, and the emission spectrum was recorded between 390 and 460nm. The dissociation constants of OBPL or NPC2 with the fluorescent probe 1-NPN was measured, and a final concentration of 2mM protein solution in 50mM Tris-HCl (pH7.4) was titrated

with aliquots of 1mM 1-NPN dissolved in methanol to final concentrations ranging from 1 to 16mM. The binding affinities of the chemicals were tested through competitive binding assays using 1-NPN as the fluorescent reporter at a concentration of 2mM, and the concentration of each competitor was varied from 0 to 32mM. The fluorescence intensities at the maximum fluorescence emission between 390 and 460nm were plotted against the free ligand concentration to determine the binding constants. The bound chemical was evaluated based on fluorescence intensity with the assumption that the protein was 100% active, with a saturation stoichiometry of 1:1 (protein: ligand). The binding curves were linearized using a Scatchard plot, and the dissociation constants of the competitors were calculated from the corresponding $IC_{50}$ values based on the following equation: $K_D = [IC_{50}]/(1+ [1\text{-NPN}]/K_{1\text{-NPN}})$, where [1-NPN] is the free concentration of 1-NPN and $K_{1\text{-NPN}}$ is the dissociation constant of the complex protein/1-NPN [48].

## Statistical analysis

Data from qPCR and EAD tests were analyzed using SPSS 17.0 (SPSS Inc., Chicago, IL, USA). ANOVA and Tukey's Honestly Significant Difference (HSD, $P = 0.05$) were used to determine differences in OBPL or NPC2 mRNA levels or whether EAD recordings were significant among different treatment groups. The t-test was employed to evaluate significant differences in OBPL or NPC2 mRNA levels in the forelegs of *H. longicornis* between different treatments.

## Results

### High *H. longicornis* burden on SFTSV-positive *Apodemus agrarius* in field trials

In our field trials, a total of 262 *Apodemus agrarius* (178 males and 84 females) were trapped and 19.84% (35 males and 17 females) were infested with immature *H. longicornis* ticks. Of the infested mice, 44 individuals (30 males and 14 females) were tested SFTSV positive and 8 (5 males and 3 females) were free of SFTSV. There were no significant difference of the tick infestation ratio and SFTSV infection rate between the sexes of *A. agrarius*. However, *A. agrarius* suffering from SFTSV infections were likely to carry more *H. longicornis* ticks than negative ones as both the overall density and intensity of infested immature *H. longicornis* on SFTSV-positive *A. agrarius* appear much higher than those on SFTSV-free mice (Pearson Chi-Square test, $\chi^2 = 37.2$, $P < 0.001$). (S1 Fig). The heavy burden of *H. longicornis* on the SFTSV-positive *A. agrarius* suggests the infestation predilection of *H. longicornis* for the infected hosts induced by SFTSV infection. And in the following, we tested our hypothesis that *H. longicornis* ticks are attracted to the characteristic host clues of SFTSV-positive mice.

### Differential volatile metabolites in fecal and urine samples of SFTSV infected *Ifnar-/-* mice

We established an untargeted GC-MS metabolomics method for fecal and urine samples of *Ifnar^-/-^* mice infected with SFTSV. We substantially extracted 672 volatile metabolites including amino acids, phenolics, indoles, dicarboxylic acids and other metabolites of microbial origin from fecal samples (S1 Data). The same method was applied to urine samples and it enabled the detection of 449 volatile metabolites as listed (S2 Data). Two technical replicates were run for each sample and only the metabolites detected in both the cases were listed, demonstrating the reproducibility of the GC-MS method. Under the criteria that VIP value >1 and $|Log_2FC| \geq 1.0$ [49], the presence of 161 volatile compounds differentially expressed in the fecal samples of the SFTSV-positive were detected (Fig 1A). Of these metabolites, 11 compounds including indoles, phenols and esters were significantly up-regulated, while another 14 compounds including bourbonene, benzaldehyde, cyclohexane and eucalyptol were down-regulated (Fig 1B). The pathways analysis performed also revealed that the differential metabolites were demonstrated close associations with their corresponding pathways although most of the metabolites were involved in more than one pathway. For example, the metabolites of tryptophan metabolism (serotonin, kynurenines, tryptamine and indolic compounds) were the most abundant. In addition, there were 55 differential metabolites in urine samples from SFTSV-positive *Ifnar^-/-^* mice, of which 26 were upregulated and 29 were

downregulated (S2 Data). The group of SFTSV-infected mice differentially expressed the inflammatory mediator-regulating pathway of the phenyl-TRP channel and phenylalanine metabolism. Among these, the significantly upregulated pathway involving phenylalanine acid, the downregulated pathways involved in eucalyptol (S2A and S2B Fig). Since indoles are recognized as quorum-sensing molecules signaling interspecies even interkingdom that meet the criteria for a perceptible host VOC cue [2–4], the present study selected two of these DVM candidates, indole or 3-methylindole, to test the potential olfactory response of *H. longicornis* and bridge the ecological gap between vector ticks and infected host.

### 3. Indolic compounds trigger the forage behavior of *H. longicornis*

**3.1 Choice test of ticks for indolic compounds.** *H. longicornis* nymphs were allowed to choose between Y-tube olfactometer arms, with one arm containing SFTSV-positive *Ifnar*$^{-/-}$ mice, healthy mice or DVM candidates (indole or 3-methylindole), and the other arm as an air control. The binomial tests showed that the ticks significantly chose the arms containing mice or candidate DVMs compared with those with either control air or solvent treatment. *H. longicornis* ticks chose healthy mice with less frequency than those to infected mice and the candidate DVMs (N = 100, $P = 0.03$). And the binomial proportion of the ticks' choices to the two candidate DVMs appeared almost identical to the choices to the SFTSV-infected mice (N = 80, $P > 0.5$). No significant difference was observed between the proportion of *H. longicornis* choosing indole or 3-methylindole (N = 40, $P > 0.5$) (Fig 2).

**Electrophysiological responses of *H. longicornis* to indole and 3-methylindole.** The stimulation from both indole and 3-methylindole could trigger the significant electrophysiological responses via Haller's organ in *H. longicornis* (S3 Fig). The active potential (AP) in olfactory circuits were shown as high as 0.18-0.28mV while the solvent control group merely yield 0.051-0.071mV AP. Compared with the control group, the differences in the EAD response values of indole and 3-methylindole to *H. longicornis* was demonstrated much higher than control with statistical significance ($F_{(2, 30)} = 231.4$ $P < 0.0001$), while no significant difference between the groups of indole and 3-methylindole ($F_{(1, 20)} = 0.005833$, $P = 0.8336$).

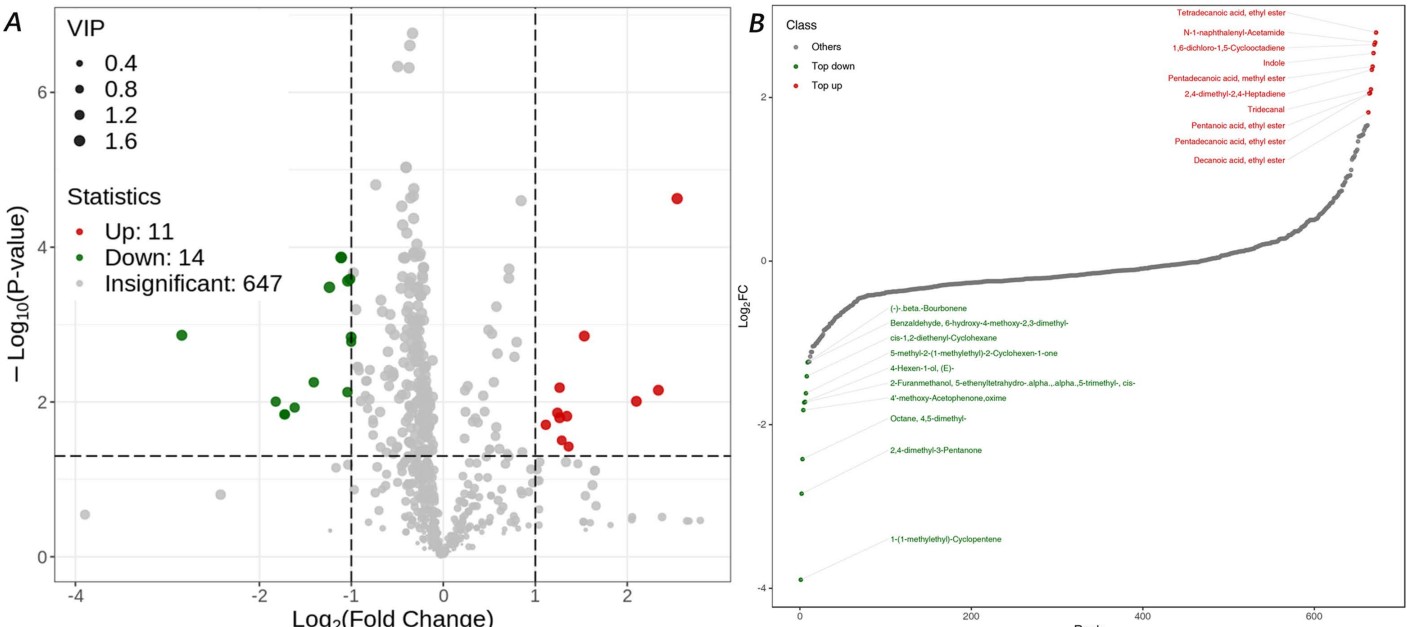

**Fig 1. Differential volatile metabolites in the fecal sample of SFTSV infected Ifnar$^{-/-}$ mice.** Panel A: volcanic diagram; Panel B: identified DVMs and their fold changes.

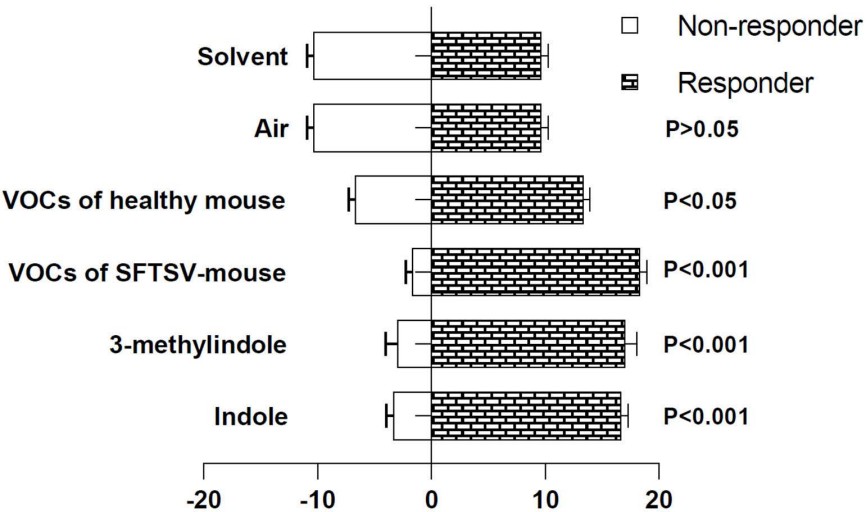

**Fig 2. Nymph *H. longicorinis* choices in Y-tubes olfactometers.** Indole (40 mg indole in a solution of 98% water and 2% ethanol); 3-methylindole (40 mg 3-methylindole in a solution of 98% water and 2% ethanol); solvent (2% ethanol, 98% water). Healthy mouse (healthy *Ifnar⁻ᐟ⁻* mouse); SFTSV-mouse (SFTSV infected *Ifnar⁻ᐟ⁻* mouse).

Moreover, there were also no significant difference of AP obtained among the different concentrations (0.1μg/mL, 1μg/mL, 10μg/mL and 100μg/mL) of indole ($F_{(4, 40)}$ = 0.9825, *P* = 0.4175) and 3-methylindole ($F_{(4, 40)}$ = 0.4958, *P* = 0.7075) respectively.

### The predicted 3D structural models and the ligand binding properties of NPC2 and OBPL in *H. longicornis*

To investigate the mechanisms of the specific attraction behavior, the olfactory genes had been retrieved from the transcriptomics data of *H. longicornis* achieved previously. As results, we failed to obtain any odorant receptor (*OR*) genes and odorant binding proteins (*OBP*) genes, which were popularly in insects of many kinds. Fortunately, a total of 8 transcripts obtained in the transcriptomics data of *H. longicornis*, which were categorized into 3 groups, NPC2, OBPL and Microplusin-like (ML), based on their sequences' identity, motif similarity (S4 Fig) and following 2D and 3D structures. With the on-line program SWISS-MODEL, the NPC2 protein structures were predicted based on the crystal structure of Bovine NPC2 (PDB ID code 1NEP) [50] due to their highest identity (95.0%) and the largest coverage of amino acid sequences (Fig 3A). The most reliable 3D model indicated that NPC2 of *H. longicornis* possesses an β-sandwich fold consisting of two orthogonally arranged β-sheets (*ca.* 30° rotation) in the helical region (S3 Table). The first β-sheet has three β-strands (β2, β4, and β6) and the second sheet has four β-strands (residues β5, β7, β8, and β9). The fourth and fifth β-strands are connected by a short half-turn α-helix (η1). There are three disulfide bonds connecting residues Cys64 in coil1 and Cys177 in β9 (S1); Cys79in α1 and Cys83 in β3 (S2); Cys130 in the η1 and Cys136 (S3). The innermost cavity in the helical region of NPC2 is formed by the following residues: Val74, Leu94, Glu95, Leu150, Val163, Trp165, and Phe178 and thought to control access of ligands to the polar-faced, water-lined internal cavity [50] (S5A and S5B Fig) although several small cavities present in the 'bottom' part of the hydrophobic core of NPC2 protein. Another cavity, formed by residues Phe107, Leu151, Val152, Leu153, Phe67, and Val180, is noticed connected to a pocket on the surface of the protein, formed by Ser116, Leu120, Phe122, Glu155, Phe156, and Pro101 residues, through a small opening topped by a 'gate' formed by Phe-156 and Phe-66. We speculate that, for larger ligand to bind, the gate would have to open up and the two β-sheets. Similarly, the 3D structure of *H. longicornis* OBPL protein were also predicted using the on-line program SWISS-MODEL based on the crystal structure of a classic OBP (PDB ID code 7NYJ) [51] from *Varroa destructor*

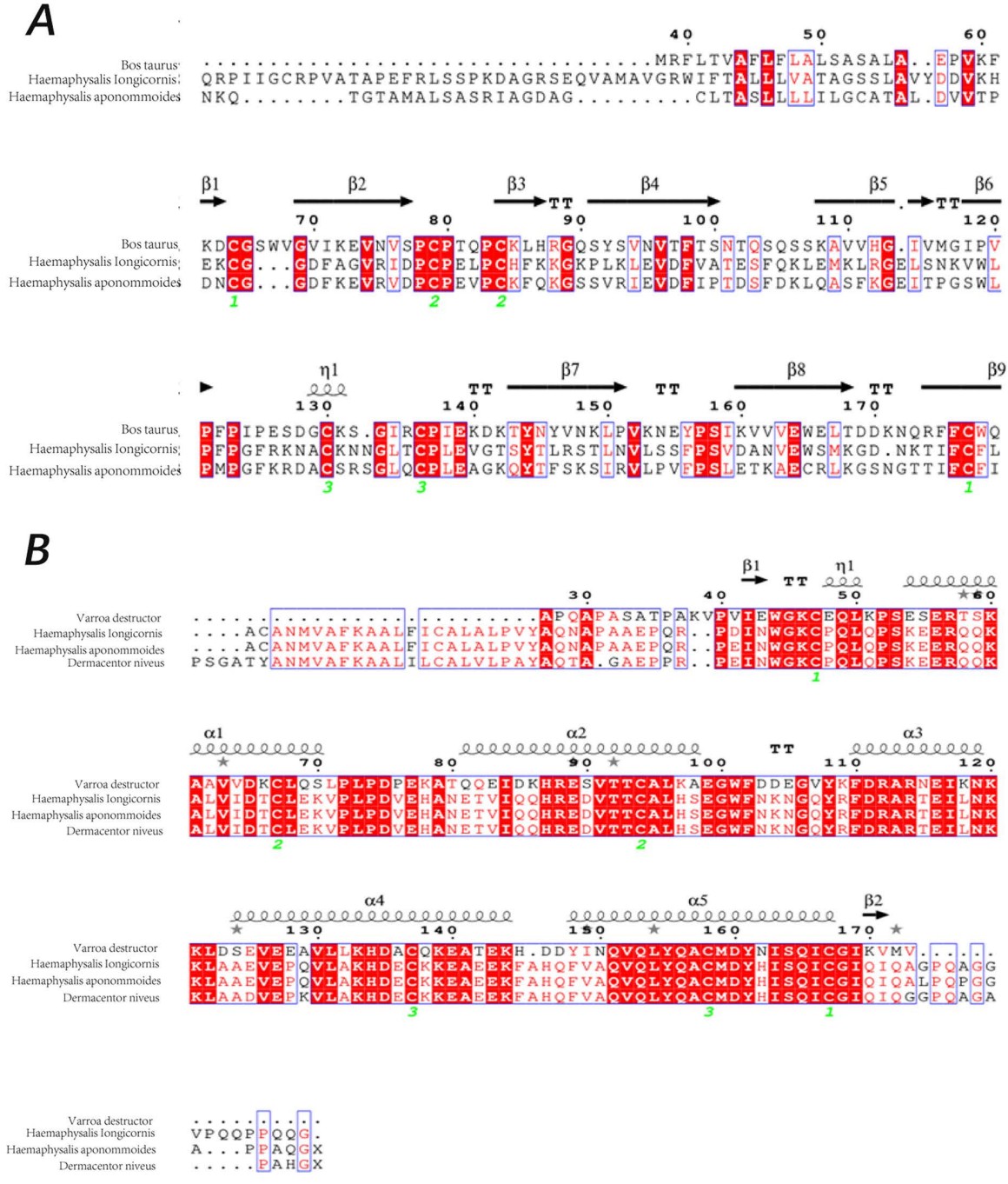

**Fig 3. The putative secondary structure of NPC2 and OBPL derived from *H. longicornis*.** Panel A: NPC2; Panel B: OBPL.

as their paired identity and coverage of amino acids sequences (Fig 3B). The scores of global model quality estimation (GMQE) and QMEAN were 0.52 and -3.65, respectively. Therefore, the predicted model of OBPL seems reasonable and reliable (S3 Table). As shown in the secondary structure, *H. longicornis* OBPL is composited of signal peptide (1–23aa), 5 α helices, 2 β folds and 2 TTs. The five α-helices locate at residues Lys48-Lys64 (α1), Glu75-Ser92 (α2), Phe104-Asn113

(α3), Glu121-Lys 138(α4), Phe143-Cys162 (α5), and three disulfide bonds connect Cys47 in coil1 and Cys167 in α5 (S1); Cys67 in α1 and Cys94 in α2 (S2); Cys137 in the α4 and Cys158 in the α5 (S3). The hydrophobic residues Phe15, Leu58, Phe59, Ala62, Val64, Leu73, Leu76, Ala79, Leu80, Ala88, Leu89, Gly92, Leu 96, Phe123, Leu124 and Ile125 from helices 1, 3, 4, 5, and loops between helices 3 and 4, and 5 form the binding cavity (S5C and S5D Fig). Interestingly, the C-terminus is pulled to the core of the protein to form part of the binding pocket wall, which can function as a "lid" for the release of ligands. The overall fold of five helices knitted together by three disulfide bridges and containing a hydrophobic binding cavity has been observed in OBPL of *H. longicornis*.

We also utilized molecular docking (SwissDock) to analyze the interactions between two putative ligands, indole and 3-methylindole with NPC2 and OBPL respectively. The results showed that all two ligands were located in the binding cavity of OBPL but interacted with different amino acids from the α-helical and motifs of loop regions in *H. longicornis* OBPL. The van der Waals interactions from hydrophobic residues and the hydrogen bonds formed by oxygen-containing functional groups and hydrogen donor residues are important linkages between the two ligands and OBPL (Fig 4A and 4B). However, the binding cavity of the two ligands to NPC2 were distinct. Both indole and 3-methylindole bind NPC2 at one surface binding cavity, instead of the buried cavity in the innermost core of OBPL. The van der Waals interactions were mainly formed between carbon-carbon double bonds and the aromatic residue Phe107. Only the indole base is predicted to overlap the center of the NPC2-binding cavity of *H. longicornis* (Fig 4C and 4D). Thus, the peculiar 3D structure of OBPL in *H. longicornis* might confer Haller's organ of *H. longicornis* with a compact pocket for odorant accommodation from indole or 3-methylindole (S5C and S5D Fig). Since the AC score in SwissDock consists of the CHARMM force field energy plus the FACTS solvation energy terms and provides an estimate of the binding free energy as a weighted sum of the polar and non-polar terms, the affinity of NPC2 (AC score -1.791514～-2.308619) and OBPL (AC score -1.696810～-2.351299) to the indoles were ranked in the same category. Due to its smaller size and weight, indole has a much greater affinity to the same target protein than 3-methylinole as shown in their SwissParam scores.

### Transcript levels of the *NPC2* and *OBPL* genes in the foreleg of *H. longicornis* responding to indole and 3-methylindole stimulation

To test the responses of *H. longicornis* to the stimulations of indole or 3-methylindole, newly molted nymphs of *H. longicornis* was allowed to exposed to indole or 3-methylindole at different concentrations and then the relative transcript level of the *NPC2* and *OBPL* genes were measured. Results indicated that the relative transcript levels of *NPC2* and *OBPL* genes changed with different patterns. The stimulations of both indole and 3-methylindole resulted in 1.2-1.4 folds change in the transcript level of *OBPL* gene under the stimulations of lower concentration (1μg/mL and 10μg/mL) (Fig 5). There was no significant difference of the relative transcript level of *OBPL* observed under the same stimulations. In comparison of *OBPL*, the *NPC2* gene reduced its transcription level to 0.6-0.8 folds change under the stimulations of higher concentration (1μg/mL and 10μg/mL) of indole or 3-methylindole (Fig 5). The up-regulated transcription of *OBPL* and down-regulated transcription of *NPC2* indicates the corresponding associations of indolergic odorants with *OBPL* in *H. longicornis*.

### Specific binding of OBPL with indole or 3-methylindole *in vitro*

The recombinant expression vector plasmid pET-28b-OBPL was successfully constructed (S6 Fig) and then transfected into host *E. coli* BL21 (DE3) and optimized host *E. coli* T7E. When induced by 1 mM IPTG under 37°C, 4h, the OBPL protein was expressed as an inclusion body with a molecular weight *ca.* 20.28kDa (Fig 6). The similar procedure was also performed for NPC2 protein and achieved its molecular weight *ca.* 15.34kDa. The purified OBPL or NPC2 (S7 Fig) proteins were submitted to the fluorescence competition binding assays of indole and 3-methylindole to validate the binding of target protein to indolergic odorants. The ligand binding characteristics of recombinant target proteins were demonstrated with competitive fluorescence binding assays with 1-NPN as competitor. The dissociation constant of the OBPL/1-NPN complex was 2.56 μM (Fig 7A) and that of the NPC2/1-NPN complex was 3.18 μM (Fig 7B). Both

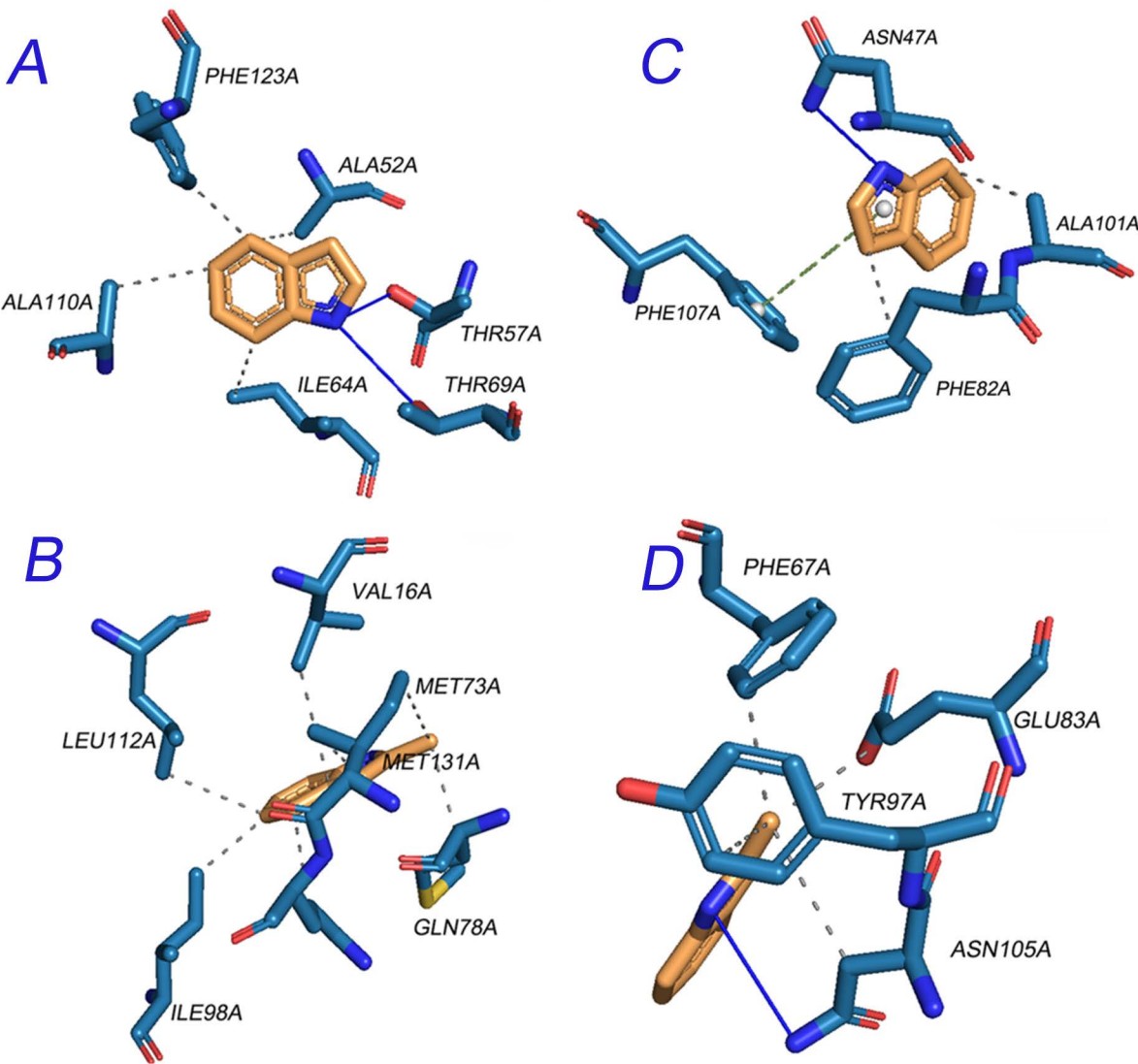

**Fig 4. The protein-ligand interactions of NPC2 and OBPL of *H. longicornis* to indole and 3-methylindole predicted with Swiss-docking.** Pane A: OBPL- Indole; Pane B: OBPL-3-methylindole; Panel C: NPC2-Indole; Panel D: NPC2-3-methylindole; (viewed and generated by PyMol 2.6.2). blue block: Protein; yellow block: Ligand; red pod: Metal Ion; dash line: Hydrophobic Interaction; blue line: Hydrogen Bond.

recombinant OBPL and NPC2 were able to bind the 2 volatile indolergic odorants, the former achieving higher affinities with Ki = 2.256 (2.043-3.135) µM for indole and Ki = 4.191 (3.357-5.277) µM for 3-methylindole (Fig 7C), compared to the moderate affinity obtained by NPC2 with Ki = 10.32 (9.540 to 13.49) µM for indole and Ki = 15.36 (11.153 to 25.85) µM for 3-methylindole (Fig 7D). The Haller's organ specific expression of the OBPL protein, and its high affinity binding to biologically active volatiles supports a possible functional role of the chemosensory proteins OBPL in the perception of general host derived odorants in *H. longicornis*. Thus, the Haller's organ-biased OBPL may mediate host recognition in the *H. longicornis* and represent new interesting targets for population control in the prevention and control strategies for tick borne diseases.

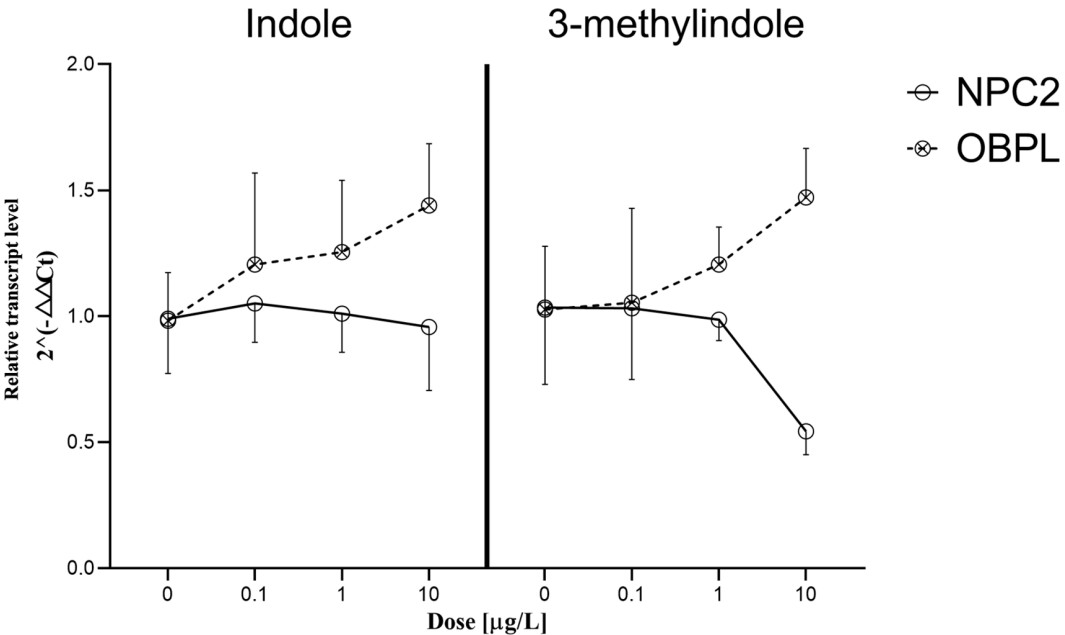

**Fig 5. The relative transcript level of NPC2 and OBPL in the forelegs of *H. longicornis* under indoles stimulations.**

## Discussion

### Upregulated indolic metabolites ameliorate pathogen-induced host damage and promote the foraging behavior of ticks for host

Research into the importance of indoles has increased dramatically, as the crosstalk between hosts and their microbiota is driven by tryptophan metabolites [52]. Various indoles have been shown to reduce bacterial biofilm formation and damage the cell surface by increasing endogenous oxidative stress, thereby inhibiting the growth of various intestinal bacteria including *Salmonella*, *Eberthella*, *Shigella*, *Escherichia* and others [53]. No wonder Bunnell *et al*. (2011) reported the significant increases in indole levels in sick hedgehogs suffering from any of 12 different ailments including lungworm (*Crenosoma striatum*), tapeworm, ringworm, breathing difficulties, hypothermia, and injury [12]. Furthermore, indolic metabolites have also been demonstrated to exert a remarkable effect in combating pathogenic viruses (Dengue, Zika and Chikungunya viruses) through drastic cell immobility and disintegration [54,55]. Viral infection triggers the stimulation of Toll-like receptors (TLRs), which induce the production of kynurenine by degrading tryptophan and activating indoleamine 2,3-dioxygenase (IDO1). IDO1 is a multifunctional enzyme and an endogenous immune checkpoint, which impairs effector T cell function, increases regulatory T cell (Treg) population and induces immune tolerance [56]. Alterations in tryptophan metabolism have been shown associated with levels of various anti-inflammatory cytokines, interleukin 6 (IL-6), tumor necrosis factor-alpha (TNFα) and other interferons (IFNs), which is produced in a rapid and transient manner in response to infection or tissue injury, and contributes to host defense by stimulating acute phase responses, haematopoiesis and immune responses [57]. Accordingly, indolic metabolism benefit the health of the host by antagonizing the proliferation of pathogens, ameliorating pathological damage and modulating host immune system. Through the complex interactions of microbiome-indoles-immune system, indoles achieve their microbiostatic functions and indicate pathological states of hosts by fostering premorbid risk factors. Therefore, the production and release of indoles in the host is an essential outcome of pathogen-host defense interactions, which could serve as a reasonable indicator of the health status of the host [58]. In the present study, the up-regulation of the tryptophan pathway and the elevated indolic metabolites

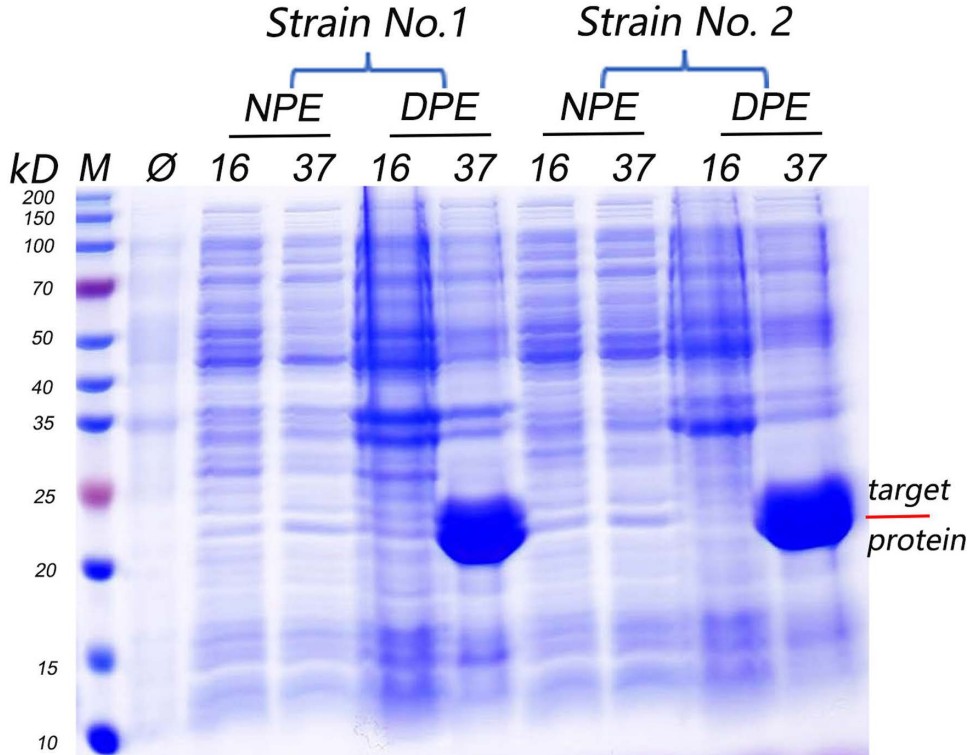

**Fig 6. The expression and purification of OBPL protein derived from *H. longicornis in vitro*.** Panel A: *in vitro* expression of OBPL in the recombined plasmid vector; NPE: supernatant DPE: inclusion protein; MW: Marker; Ø: host bacteria (without induction); strain No.1: BL21(DE3) strain; strain No.2: T7E strain.

in the SFTSV-infected mice reconfirmed that the production of indoles can be induced by viral infections and that the increased indoles potentially benefit the host with a remarkable amelioration of virological damage. More importantly, our study provides strong evidence that ectoparasitic ticks acutely sense the metabolic changes resulting from the virus-host interaction and then simultaneously trigger or promote the tick's foraging behavior for the particular host suffering from viral infection. Among these, the perception of elevated indolic metabolites may be a crucial process to charge the tick's foraging behavior, although the detailed signaling and response processes remain to be elucidated.

## Smell of indoles guide tick foraging behavior for infected host

Physiologically, ticks and other blood-feeding arthropods have overlapping receptive fields for many aromatics and heterocyclics [59]. Among these, indole and 3-methylindole, the derivatives from bacterial degradation of tryptophan, were also known to mediate long-range attraction in several culicine and anopheline species [60,61]. The former aids the location of blood host by host recognition, while the latter acts as an oviposition attractant. Indeed, indoles elicit strong physiological responses in adult antennal trichoid sensilla of *An. gambiae* [62,63], and olfactory receptor neurons (ORNs) activation in *Ae. aegypti* [64], *C. quinquefasciatus* [60], and *C. tarsalis* [65]. In addition, 4-methyl-phenol (4-MP or p-cresol), one aromatic compound found in human sweat and hay infusion [65], plays an important role as an attractant for tick's blood feeding and *Aedes*, *Culex*, and *Anopheles* mosquitoes' oviposition [60,61,64,65]. However, the function of the indoles to tick's host foraging, toxin and predator avoidance, egg deposition, and mate selection has yet to be determined. In the present studies, the increased level of indolergic metabolites in the infected host resulted

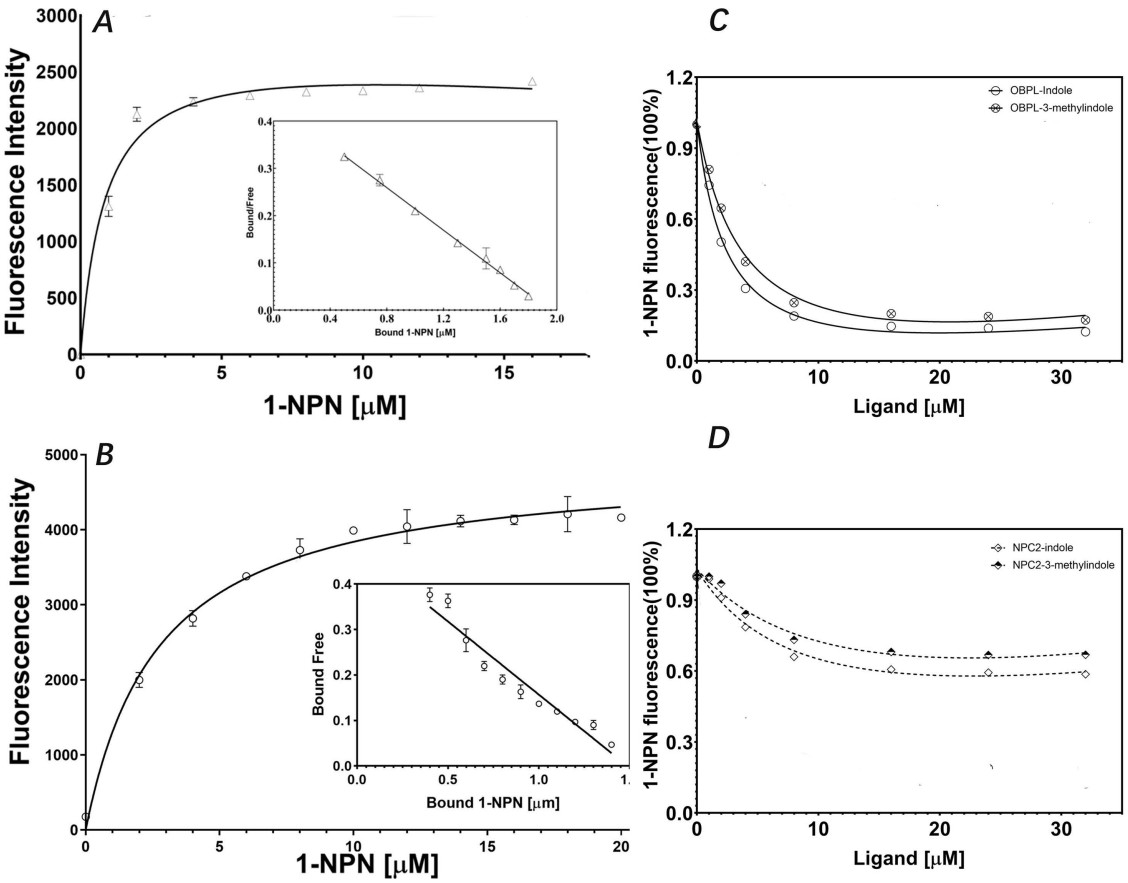

**Fig 7. The competitive fluorescence binding assays on OBPL/NPC2 to indoles.** Panel A: The dissolution constant of OBPL to 1-NPN; Panel B: the dissolution constant of NPC2 to 1-NPN; Panel C: The binding activity of OBPL to indole and 3-methylindole; Panel D: The binding activity of NPC2 to indole and 3-methylindole.

in high levels of *H. longicornis* infestation, suggesting the possible attractiveness of indoles to host-seeking ticks, which may partially explain the interesting phenomenon that higher tick burdens occur on the sick hedgehogs reported by Bunnell *et al.* (2011). Although in their later experiment, the addition of indole to the feces of healthy hedgehogs did not attract the ticks tested as expected [12], the announced proposal that ticks may choose their host based on indole linked to the host's health status, as the attractiveness of these indolergic metabolites to ticks may be masked or inhibited by some unidentified substances and their potential synergistic effects. Most interestingly, indolergic metabolites were also found to attract or repellent to various vector insects with several odorant receptors (ORs) and odorant binding proteins (OBPs) involved, including CquiOBP1 (*Culex quinquefasciatus*), CquiOR2, CquiOR10, AalbOR10 (*Aedes albopictus*), AgamOR10 (*An. gambiae*), and AsinOR10 (*An. sinensis*) [66]. These include olfactory indolergic receptors, commonly found in mosquitoes and other highly chemo-sensitive insects, which allow this large and diverse taxonomic group to exploit a dazzling array of ecological niches. However, the known olfactory receptors are only present in insects, but not in Arachnida. Thus, the identification of OBP analogous and their appropriated ligands are considered crucial for the chemosensory behavior of the Arachnida species, since they can bind chemical cues and transport them to the appropriate receptors via the sensillum lymph [65]. To the best of our knowledge, efforts afford to identify olfactory associated proteins failed to yield any OBP in Arachnida species instead of NPC2 protein, Microplusin-like (ML)

protein and OBPL protein, despite that large numbers of OBPs have been identified in hemipteran insect species [66]. Nowadays, as small soluble proteins belonging to the myeloid differentiation factor 2 (MD-2) related lipid-recognition protein family, NPC2 proteins are functionally similar to odorant binding proteins (OBPs) and involved in chemical communication in arthropods [67,68]. Moreover, the typical β-barrel or sandwich structure of the NPC2 protein, consisting of eight β-helices and a short α-helical segment, can bind the signaling chemicals extensively [69], suggesting a potential function as an olfactory recognition protein in the tick species. Till date, NPC2 has been found in *Ixodes ricinus* [70], *Amblyomma americanum* [71], *I. scapularis* [72], *H. longicornis* [40], *Varroa destructor* [73], *Pardosa pseudoannulata* [74], *Camponotus japonicus* [69], *Helicoverpa armigera* [67] and others arthropods [75]. However, the exact odor binding activities and their corresponding upstream and downstream signal transduction molecules of the NPC2 proteins in ticks are still unclear. Similar to NPC2, ML proteins also have been annotated from a foreleg transcriptome analysis [72] of *I. scapularis* and a proteome analysis of *A. americanum* [71], as well as large-scale, tick genomic studies [76]. Although they were exclusive and biased expressed in Haller's organ, a single, comparable ligand-binding site might be insufficient to elucidate the olfactory chemo-sensation role of ML in ticks, although relevant functional experiments have yet to be performed. Till to now, only two foreleg-biased OBPL proteins, OBPL-1 (19 aa; 118 aa) and OBPL-2 (24 aa; 150 aa), that resemble insect odorant-binding proteins were discovered in a chemosensory proteome of *A. americanum* [71]. Concerning OBPL, a classic OBPL from *H. longicornis* was fortunately identified and its potential binding properties with indolic ligands were analyzed in our study. Our current functional analysis based on amino acid sequences and three-dimensional folding suggests that the classic OBPL comply with the binding protein criteria for odorant ligands proposed by Pelosi *et al.* [68]. (1) The classic OBPL presents a signal peptide revealing their secretory nature. (2) the classic OBPLs are made of five α-helical domains connected by short unstructured loops and knitted together in a compact and stable structure by three disulfide bridges [77]. (3) A comparable, central hydrophobic binding cavity predicted enable the classic OBPL with binding activity with appropriated odorant ligands. Together with the relative high expression level of the *OBPL* in forelegs of *H. longicornis*, we come to conclude that the classic OBPLs might function as one odorant binding protein to recognize and transport volatile odorant compounds. Indeed, our competitive fluorescent binding assays also confirmed a specific binding preference of the classic OBPLs for indoles, indole and 3-methylindole, with high affinities, which play a notable role in *H. longicornis* olfactory system.

## Control strategies inspired by tick olfactory ecology

Ticks display robust olfactory-driven behaviors as its olfactory receptor neurons (ORNs) that innervate the sensilla in the Haller's organ are able to detect various behavior-modifying semiochemicals extensively [78]. Most of these semiochemicals are distinct but limited range of volatiles from host animals, which can elicit specific behaviors such as host foraging or natural enemy avoidance [79]. For example, *meth*-cresol shows the strongest electrophysiological response and clearly attracts ticks to feed, while *ortho*-methylphenol enables ticks to efficiently find a mating partner [79]. Using the delicate and intricate olfactory mechanism, ticks bridge the gap between different pathogens and their hosts, and efficiently spread diverse tick-borne pathogens to humans. Till date, much researches have focused on the determine of tick's odorant receptors (ORs), OBPs, and their appropriate volatiles, which will lead to the development of odor baits to be used in tick management programs [79]. However, little work has been done to exploit olfactory links between vector ticks and infected hosts under conditions of pathological damage. Since characteristic host odors are usually produced and released during infection with specific microbiota or set of pathogens, a full understanding of the characteristics of tick-borne disease transmission pathways or cycles would help develop scientific control strategies based on characteristic pathogen-driven host odors. Inspired by the close links between vector arthropods and hosts infected with various pathogens as described above, we used olfactory ecology theory to test the semiochemical links between vector ticks and SFTSV-infected hosts. Our studies, for the first time in the sight of olfactory ecology, have successfully revealed the olfactory link between SFTSV infected host and host seeking *H. longicornis*. Although our results are limited and infantile, the efficient olfactory

links between *H. longicornis* and SFTSV infected mice may not be limited within indolic volatiles. Other differential metabolites should not be excluded bridging the gaps between the vector tick and SFTSV infected hosts. For which, more efforts in detail should be determined in the future. Nevertheless, our preliminary results would pave a way to device a newly control strategy for tick borne disease based on olfactory ecological theory.

## Supporting information

**S1 Table. Software used in the present study.**
(DOCX)

**S2 Table. Primers and probes for SFTSV screening.**
(DOCX)

**S3 Table. 3D homology modeling parameters for NPC2 and OBPL in Swiss-model.**
(DOCX)

**S1 Fig. *H. longicornis* burden on the SFTSV infected *Apodemous agrarius*.**
(TIF)

**S2 Fig. Differential volatile metabolites in the urine sample of SFTSV infected *infar*^/- mice.** Panel A: volcanic diagram of the identified DVMs; Panel B: identified DVMs and their fold changes.
(TIF)

**S3 Fig. EAD records of Haller's organ of *H. longicornis* under the stimulations of indole or 3-methylindole.** Panel A: indole under different concentrations; Panel B: 3-methylindole under different concentrations.
(TIF)

**S4 Fig. Phylogenetic tree of NPC2 and OBPL among the different taxa of Arachnid species.** Panel A: NPC2 (HaeL NPC2 accession no. PV029724); Panel B: OBPL (HaeL OBPL accession no. PV029725).
(TIF)

**S5 Fig. Molecular docking of NPC2 and OBPL of *H. longicornis* to indole and 3-methylindole predicted with Swiss-docking.** Panel A: NPC2- Indole; Pane B: NPC2–3-methylindole; Pane C: OBPL-Indole; Panel D: OBPL-3-methylindole).
(TIF)

**S6 Fig. The identification of recombined expression plasmid of pET-28b-OBPL.** Panel A: the recombined expression plasmid of pET-28b-OBPL; Panel B: the electrophoresis panel of pET-28b-OBPL digested. M: molecular marker, 1–2, the recombined expression plasmid; 3: pVK type 3, blank plasmid pET-28b and target OBPL gene.
(TIF)

**S7 Fig. The purified NPC2 of *H. longicornis* expressed in *Escherichia coli* BL21(DE3) *in vitro*.** M: Marker; IN: unpurified mixtures; FT: Fluid flow through empty column; W: PBS buffer; E: 300 mM Mimidazole elution.
(TIF)

**S1 Data. Differential volatile metabolites identified in the fecal samples of SFTSV infected *infar*^/- mice.**
(XLSX)

**S2 Data. Differential volatile metabolites identified in the urine samples of SFTSV infected *infar*^/- mice.**
(XLSX)

## Acknowledgments

We are grateful to Prof. Wuchun Cao in the State Key Laboratory of Pathogen and Biosecurity, Academy of Military Medical Sciences of People's Republic of China for his helpful discussions and suggestive review on the article.

## Author contributions

**Conceptualization:** Zhitong Liu, Yi Sun, Jiafu Jiang.

**Data curation:** Zhitong Liu, Hao Feng, Hong Zhang.

**Formal analysis:** Zhitong Liu, Hao Feng, Yi Sun.

**Funding acquisition:** Yi Sun.

**Investigation:** Zhitong Liu, Jiahong Wu.

**Methodology:** Zhitong Liu, Xiaohe Liu, Yi Sun, Chunxiao Li.

**Project administration:** Yi Sun.

**Resources:** Yi Sun, Jiahong Wu, Jiafu Jiang.

**Software:** Zhitong Liu, Xiaohe Liu, Bin Wu, Yi Sun.

**Supervision:** Yi Sun, Chunxiao Li.

**Validation:** Zhitong Liu, Yi Sun, Chunxiao Li.

**Visualization:** Zhitong Liu, Yi Sun.

**Writing – original draft:** Zhitong Liu, Hao Feng, Xiaohe Liu.

**Writing – review & editing:** Zhitong Liu, Hao Feng, Yi Sun, Jiahong Wu, Jiafu Jiang.

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
