## [Decision Letter · Decision Letter 0]

Response to Reviewers
Revised Manuscript with Track Changes
Manuscript

Yong Qi

Academic Editor

Shaden Kamhawi

co-Editor-in-Chief

Paul Brindley

co-Editor-in-Chief

**Journal Requirements:**

At this stage, the following Authors/Authors require contributions: Chunxiao Li. Please ensure that the full contributions of each author are acknowledged in the "Add/Edit/Remove Authors" section of our submission form.

- © on page: 29.

- ® on page: 22.

5) We have noticed that you have uploaded Supporting Information files, but you have not included a list of legends. Please add a full list of legends for your Supporting Information files after the references list.

6) We note that your Data Availability Statement is currently as follows: "The transcriptomic and metabolomic datasets generated during the current study are available from the corresponding author on reasonable request.". Please confirm at this time whether or not your submission contains all raw data required to replicate the results of your study. Authors must share the “minimal data set” for their submission. PLOS defines the minimal data set to consist of the data required to replicate all study findings reported in the article, as well as related metadata and methods (https://journals.plos.org/plosone/s/data-availability#loc-minimal-data-set-definition).

- The points extracted from images for analysis..

7) Please amend your detailed Financial Disclosure statement. This is published with the article. It must therefore be completed in full sentences and contain the exact wording you wish to be published. Please ensure that the funders and grant numbers match between the Financial Disclosure field and the Funding Information tab in your submission form. Note that the funders must be provided in the same order in both places as well.

**Reviewers' comments:**

**Key Review Criteria Required for Acceptance?**

**Methods**

-Are the objectives of the study clearly articulated with a clear testable hypothesis stated?

-Is the study design appropriate to address the stated objectives?

-Is the population clearly described and appropriate for the hypothesis being tested?

-Is the sample size sufficient to ensure adequate power to address the hypothesis being tested?

-Were correct statistical analysis used to support conclusions?

-Are there concerns about ethical or regulatory requirements being met?

Reviewer #1: (No Response)

Reviewer #2: Yes. The objectives of the study clearly articulated with a clear testable hypothesis stated.

No. The study design should be improved to address the stated objectives and make the results more convincing.

Yes. The population clearly described and appropriate for the hypothesis being tested.

Yes. The sample size is sufficient to ensure adequate power to address the hypothesis being tested. but the authors should carefully specify the sample size, especially in "Choice tests" and "Potent electrophysiological responses..." sections.

Yes. Correct statistical analysis were used to support conclusions.

No. The authors had a well-described Ethics in their manuscript.

Reviewer #3: The objectives of the study clearly articulated with a clear testable hypothesis stated, the study design appropriate to address the stated objectives, the population clearly described and appropriate for the hypothesis being tested, the sample size sufficient to ensure adequate power to address the hypothesis being tested, correct statistical analysis used to support conclusions, there aren't concerns about ethical or regulatory requirements being met.

**Results**

-Does the analysis presented match the analysis plan?

-Are the results clearly and completely presented?

-Are the figures (Tables, Images) of sufficient quality for clarity?

Reviewer #1: (No Response)

Reviewer #2: Yes. The presented analysis match the analysis plan.

Yes. The results are clearly and completely presented.

Yes. The figures (Tables, Images) are of sufficient quality for clarity.

Reviewer #3: The analysis presented match the analysis plan, the results clearly and completely presented, and the figures (Tables, Images) of sufficient quality for clarity.

Strengths:

Novelty: The study provides new insights into the olfactory mechanisms underlying tick attraction to infected hosts. This is crucial for understanding tick-borne disease transmission.

Rigorous Methodology: The research employs a combination of field studies, laboratory experiments, and molecular analyses to support its conclusions.

Clear Presentation: The paper is generally well-written and easy to follow. The figures and tables are informative and effectively illustrate the key findings.

Areas for Improvement:

Further Exploration of NPC2: While the study focuses on OBPL, further investigation into the role of NPC2 in tick olfaction is warranted. Its downregulation in response to indole stimulation raises interesting questions about its potential function.

Specificity of Indole Attraction: It would be valuable to explore whether the observed attraction to indoles is specific to SFTSV infection or if it extends to other pathogens that may also alter host odor profiles.

Ecological Implications: A more in-depth discussion of the ecological implications of these findings would be beneficial. How might this knowledge be used to develop novel tick control strategies?

**Conclusions**

-Are the conclusions supported by the data presented?

-Are the limitations of analysis clearly described?

-Do the authors discuss how these data can be helpful to advance our understanding of the topic under study?

-Is public health relevance addressed?

Reviewer #1: (No Response)

Reviewer #2: Yes. The conclusions are supported by the data presented.

Yes. The limitations of analysis are clearly described.

Yes. The authors did discuss how these data can be helpful to advance our understanding of the topic under study.

No. It seems like that this manuscript does not refer to public health relevance.

Reviewer #3: The research provides strong evidence that SFTSV infection in mice alters their odor profile, making them more attractive to Haemaphysalis longicornis ticks.

Indole and 3-methylindole appear to be key components of this "infection scent."

The OBPL protein plays a crucial role in the tick's perception of these indole odors.

This finding suggests potential strategies for tick control by manipulating host odor or tick olfactory receptors.

This research sheds light on the complex interactions between ticks, hosts, and pathogens. Understanding these mechanisms could lead to novel methods for preventing tick-borne diseases like SFTSV.

**Editorial and Data Presentation Modifications?**

Reviewer #1: (No Response)

Reviewer #2: 1) Authors should include page numbers and use continuous line numbers to make it easier to locate the sentence.

2) Page 2 Line 3: change “tick’s” to “ticks’”

3) Page 2 Line 7: change “faces” to “fecal”

4) Ensure consistency in abbreviations throughout the whole manuscript (e.g. Hae. longicorinis or H. longicorinis)

5) Page 4 Line 12: make the sentence easier to understand (Thus,…)

6) Page4 Line 15-17: is there any reference?

7) Page5 Line 16: “for the presence of” or “in the presence of”?

8) At the end of the second part of the discussion, where does figure 9 come from? “…(Figure 9), we come to…”

9) In Supplemental Figure 2, is panel A totally same with panel B?

10) Ensure consistency "Supplemental Figure SNo." or "Supplemental Figure No." throughout the whole manuscript

Reviewer #3: Good

**Summary and General Comments**

Reviewer #1: The contents described in this paper contain interesting results on disease-carrying hard ticks. However, there are basic errors throughout the paper. The following contents seem to need to be corrected.

1. Abbreviations should be sereioly cheked throughout the paper. It is necessary to check whether Hae. longicornis is the correct abbreviation for Haemaphysalis longicornis. It is generally written as H. longicornis in many thesis, and two abbreviations are used interchangeably in this paper (Hae. longicornis & H. longicornis). Italics and abbreviations need to be unified, including the scientific names described in the data and data legend.

2. There are various typos throughout the paper. Various typos were found, such as writing feces as faces. The overall content of the paper needs to be carefully checked and typos and grammatical errors need to be corrected.

3. The overall resolution of the data is poor. It would be desirable to revise this.

4. The Discussion section is somewhat long and the content is redundant. It would be better to concisely describe the key mechanisms and the possible contributions of the research results. In this regard, it would not be a bad idea to add visual materials that clearly summarize the experimental process and results.

Reviewer #2: The author identified two characteristic indolic scents from hosts using animal models infected with SFTSV. This research highlights the olfactory connection between SFTSV-infected hosts and the host-seeking behavior of Haemaphysalis longicornis through a series of biological experiments. This intriguing work may offer a new strategy for controlling tick-borne diseases. However, I have identified a few major and minor issues with this paper.

The authors devoted significant space to discussing the relationship between microbiota and odor; however, this was not substantiated by the actual research findings. It is recommended to remove these redundant descriptions to enhance the conciseness of the article.

The authors did not adequately explain why indole or 3-methylindole was chosen as the focus of their research. After analyzing the data from the GC-MS, it appears that these two substances did not show the most significant increase. Why not choose other substances as subjects? Why were no other substances set as controls during the experiment? Did the author attempt to use the method described in the reference (López-López N et al., J Med Entomol. 2023;60(3):432-442. doi:10.1093/jme/tjad019) to evaluate synthetic compounds within alone, or in binary, tertiary, or quaternary mixtures? How did they determine whether indole or 3-methylindole has the best attraction effect? Merely stating that indoles are recognized as interspecies and inter-kingdom signal molecules is insufficient evidence.

In the second part of the results, did the author test both indole and 3-methylindole to determine whether they have synergistic, additive, or antagonistic effects?

In the fourth part of the results, it is insufficient to demonstrate the correlation between small molecule compounds and target genes based solely on transcriptional level validation. Additionally, from a data perspective, the changes observed are not very significant, ranging from 1.2 to 1.4-fold or as low as 0.6 to 0.8-fold. Is there any relevant literature that discusses the potential relationship between non-significant transcriptional expression changes in targets and compound interactions? If not, these findings could be included in supplementary materials or further explored through other approaches, such as examining translational levels or enzyme activity, etc.

On the other hand, if the conclusion in the fourth part of the results is valid, then NPC2 protein should be included as a control in the fifth part of the results. Have the authors attempted to validate the binding of the molecule and the protein using Microscale Thermophoresis (MST) experiments? These experiments provide a more accurate and credible assessment of protein-odor molecule interactions in an environment that is closer to the original conditions.

In the reference (Bunnell T, et al., J Chem Ecol. 2011;37(4):340-347. doi:10.1007/s10886-011-9936-1), an interesting phenomenon is reported. Specifically: 1) Sick hedgehogs tended to have elevated levels of indole in their feces. 2) Ticks were attracted to indole when given the option between indole and a solvent control. 3) Fecal matter from healthy hedgehogs, even with the addition of indole, did not attract ticks. However, the authors of this manuscript did not address the relationship between the aforementioned phenomenon and the experimental results presented in this study.

Reviewer #3: The research provides strong evidence that SFTSV infection in mice alters their odor profile, making them more attractive to Haemaphysalis longicornis ticks.

Indole and 3-methylindole appear to be key components of this "infection scent."

The OBPL protein plays a crucial role in the tick's perception of these indole odors.

This finding suggests potential strategies for tick control by manipulating host odor or tick olfactory receptors.

This research sheds light on the complex interactions between ticks, hosts, and pathogens. Understanding these mechanisms could lead to novel methods for preventing tick-borne diseases like SFTSV.

PLOS authors have the option to publish the peer review history of their article (what does this mean? ). If published, this will include your full peer review and any attached files.

**Do you want your identity to be public for this peer review?** For information about this choice, including consent withdrawal, please see our Privacy Policy .

Reviewer #1: No

Reviewer #2: No

Reviewer #3: **Yes: ** Nabil Abo Kaf

**Figure resubmission:****Reproducibility:** To enhance the reproducibility of your results, we recommend that authors of applicable studies deposit laboratory protocols in protocols.io, where a protocol can be assigned its own identifier (DOI) such that it can be cited independently in the future. Additionally, PLOS ONE offers an option to publish peer-reviewed clinical study protocols. Read more information on sharing protocols at https://plos.org/protocols?utm_medium=editorial-email&utm_source=authorletters&utm_campaign=protocols

---

## [Decision Letter · Decision Letter 1]

Dear Dr. Yi,

Response to Reviewers
Revised Manuscript with Track Changes
Manuscript

Shaden Kamhawi

co-Editor-in-Chief

Paul Brindley

co-Editor-in-Chief

**Journal Requirements:**

1) Some material included in your submission may be copyrighted. According to PLOS’s copyright policy, authors who use figures or other material (e.g., graphics, clipart, maps) from another author or copyright holder must demonstrate or obtain permission to publish this material under the Creative Commons Attribution 4.0 International (CC BY 4.0) License used by PLOS journals. Please closely review the details of PLOS’s copyright requirements here: PLOS Licenses and Copyright. If you need to request permissions from a copyright holder, you may use PLOS's Copyright Content Permission form.

Potential Copyright Issues:

[Figure 8]: Thank you for previously providing the copyright information for Figure 8. The GDP license is too restrictive compared to the PLOS CCBY 4.0 license, allowing for commercial reuse with attribution. Please either replace the images with CCBY 4.0 images or Public Domain or seek permission from the GDP team.

**Reviewers' comments:**

**Key Review Criteria Required for Acceptance?**

**Methods**

-Are the objectives of the study clearly articulated with a clear testable hypothesis stated?

-Is the study design appropriate to address the stated objectives?

-Is the population clearly described and appropriate for the hypothesis being tested?

-Is the sample size sufficient to ensure adequate power to address the hypothesis being tested?

-Were correct statistical analysis used to support conclusions?

-Are there concerns about ethical or regulatory requirements being met?

Reviewer #2: The author applied reasonable research methods to to verify their scientific hypotheses.

Reviewer #3: (No Response)

**Results**

-Does the analysis presented match the analysis plan?

-Are the results clearly and completely presented?

-Are the figures (Tables, Images) of sufficient quality for clarity?

Reviewer #2: The results clearly and completely presented.

Reviewer #3: (No Response)

**Conclusions**

-Are the conclusions supported by the data presented?

-Are the limitations of analysis clearly described?

-Do the authors discuss how these data can be helpful to advance our understanding of the topic under study?

-Is public health relevance addressed?

Reviewer #2: The authors had a good discussion on the research results, but the focus of the discussion still needs to be carefully considered.

Reviewer #3: (No Response)

**Editorial and Data Presentation Modifications?**

Reviewer #2: 1) line 92, 95: strategists or strategies?

2) line 113: change to “sebaceous”

3) line 139-140: First thing, this sentence did not elaborate on the relationship between VOCs and attracting vectors, plus, incorrect citation of literature. Is the content from Reference No. 25 actually? It seems that the citation order of most references is messy. Please ensure the correctness and citation order of the literature references.

4) line 144: what dose “intense immature” mean?

5) line 176: change "26 °C" to "26°C". Please ensure uniform format of measurement units throughout the whole manuscript. Pay attention spaces between numbers and units.

6) line 187, 215, 216, 219…: same suggestion as above.

7) line 273: what dose “EAG” mean?

8) line 463: provide the full name: Microplusin-like (ML)

9) line 466: change “Figure 4” to “Figure 3”

10) line 524-525: under the stimulations of lower concentration

11) line 934: Panel B, Panel C

12) line 940-941: Italic “in vitro”

13) line 947: The dissolution

14) line 974: change to “1-2”; “3:”

15) for Figure 4, authors should show more details in the pictures, i.e. key amino acid binding sites.

Reviewer #3: (No Response)

**Summary and General Comments**

Reviewer #2: Before acceptence, I still have identified a few minor issues with this paper.

1) This article focuses on the negative impact of tick-borne virus infection on tick feeding behavior. However, in the second paragraph of the discussion, this section is specifically devoted to explaining the benefits of indole metabolism on host health from an immune perspective, which confuses readers. Furthermore, in Figure 8, authors should emphasize current experimentally process and results.

2) Many typos, misuse of punctuation marks, incorrect citation order of references and inconsistent unit formats were still found. The whole manuscript including supplemental materials needs to be carefully checked.

Reviewer #3: (No Response)

PLOS authors have the option to publish the peer review history of their article (what does this mean? ). If published, this will include your full peer review and any attached files.

**Do you want your identity to be public for this peer review?** For information about this choice, including consent withdrawal, please see our Privacy Policy .

Reviewer #2: No

Reviewer #3: **Yes: ** Nabil Abo Kaf

**Figure resubmission:****Reproducibility:** To enhance the reproducibility of your results, we recommend that authors of applicable studies deposit laboratory protocols in protocols.io, where a protocol can be assigned its own identifier (DOI) such that it can be cited independently in the future. Additionally, PLOS ONE offers an option to publish peer-reviewed clinical study protocols. Read more information on sharing protocols at https://plos.org/protocols?utm_medium=editorial-email&utm_source=authorletters&utm_campaign=protocols

---

## [Editor Report · Decision Letter 2]

Dear Prof. Yi,

We are pleased to inform you that your manuscript 'SMELLY COMMUNICATION BETWEEN HAEMAPHYSALIS LONGICORNIS AND INFECTED HOSTS WITH INDOLIC ODORANTS: A CASE FROM SEVERE FEVER WITH THROMBOCYTOPENIA SYNDROME VIRUS' has been provisionally accepted for publication in PLOS Neglected Tropical Diseases.

Best regards,

Yong Qi

Academic Editor

Amy Morrison

Section Editor

Shaden Kamhawi

co-Editor-in-Chief

Paul Brindley

co-Editor-in-Chief

---

## [Editor Report · Acceptance letter]

Dear Prof. Sun,

We are delighted to inform you that your manuscript, "SMELLY COMMUNICATION BETWEEN HAEMAPHYSALIS LONGICORNIS AND INFECTED HOSTS WITH INDOLIC ODORANTS: A CASE FROM SEVERE FEVER WITH THROMBOCYTOPENIA SYNDROME VIRUS," has been formally accepted for publication in PLOS Neglected Tropical Diseases.

Best regards,

Shaden Kamhawi

co-Editor-in-Chief

Paul Brindley

co-Editor-in-Chief
